# Influence of High Hydrostatic Pressure on the Identification of *Lactobacillus* by MALDI-TOF MS- Preliminary Study

**DOI:** 10.3390/microorganisms8060813

**Published:** 2020-05-28

**Authors:** Joanna Bucka-Kolendo, Barbara Sokołowska, Stanisław Winiarczyk

**Affiliations:** 1Department of Microbiology, Institute of Agriculture and Food Biotechnology, 02-532 Warsaw, Poland; barbara.sokolowska@ibprs.pl; 2Institute of High Pressure Physics, Polish Academy of Sciences, 01-142 Warsaw, Poland; 3Faculty of Veterinary Medicine, University of Life Sciences in Lublin, 20-612 Lublin, Poland; genp53@interia.pl

**Keywords:** *Lactobacillus*, lactic acid bacteria, high hydrostatic pressure, MALDI-TOF MS, matrix-assisted laser desorption ionization time-of-flight mass spectrometry

## Abstract

In the present study, we assessed the ability of MALDI-TOF MS (matrix-assisted laser desorption ionization time-of-flight mass spectrometry) to identify microbial strains subjected to high hydrostatic pressure (HHP) as a stress factor. Protein changes induced by HHP can affect the identification of microorganisms when the identification technique is based on the protein profile. We evaluated two methods, namely MALDI-TOF MS and 16S rDNA sequencing, as a valuable tool to identify *Lactobacillus* species isolated from spoiled food, juices and beers. The data obtained from the protein mass fingerprint analysis of some of the lactobacilli strains showed differences in unpressured and pressured mass spectrum profiles (MSPs), which influenced the results of the identification. Four out of 13 strains (30%) showed different MSP results for unpressured and pressured samples and these results did not overlap with the 16S rDNA identification results. The 16S rDNA sequencing method revealed that five unpressured strains (38%) and four pressured strains (40%) were identified correctly by MALDI-TOF MS. Both methods showed compatible results in 38% of unpressured strains and in 30% of pressured strains. Stress factors, cultivation methods or the natural environment from which the bacteria were derived can affect their protein profile and thus change the mass spectrum. It is necessary to expand the database with a wide range of mass spectra dedicated to a high-throughput study of the microorganisms derived from different environments.

## 1. Introduction

High hydrostatic pressure (HHP) treatment is considered as a promising nonthermal pasteurization method that inactivates foodborne pathogens and spoilage microorganisms [1]. HHP enables to reduce the count of spoilage-causing microflora while keeping the sensory and nutritional quality of the treated product virtually intact. Current studies on the response of lactic acid bacteria (LAB) to stress caused by HHP are focused on the changes in the structure, metabolism, growth and viability of the cells [2,3,4,5]. As reported previously [6], HHP can influence bacterial proteome and induce protein conformational changes. The increase in the pressure can lead to the reduction in the permeability of the cell membrane and change in the conformation of proteins. This might affect their properties and functions or might denature them completely [7,8]. Cell inactivation by HHP is mainly related to protein denaturation, which can cause enzyme inactivation and cell protein agglomeration [2]. LAB under stress conditions can activate their defense mechanisms, manifesting in stress-related protein secretion. For a better understanding of the relationship between the molecular basis of the spoiling abilities of *Lactobacillus* and the microorganisms itself, it is crucial to link the function of the proteins with the molecular mechanisms of the cellular stress reaction. The strains collected from environmental samples have to be identified in a quick and reliable manner, as it is important for the manufacturers to respond rapidly and prevent further product deterioration. Higher taxonomic resolution and identification of bacteria below the species level, i.e., bacterial subtyping, are needed to ensure food safety and meet the increasing requirements of the food industry [9]. Such methods allow to track specific foodborne pathogens through the entire production process and to identify critical points where contamination/unwanted growth occurs.

The most popular genotyping methods used to identify bacteria are those based on polymerase chain reaction (PCR), especially those involving the sequencing of the 16S rDNA region. This area is a component of the 30S small subunit of prokaryotic ribosomes and is characterized by a slow rate of evolution; it encodes the 16S rRNA gene that can provide species-specific signature sequences useful in the identification of bacteria and the determination of their phylogenetic position [10]. The 16S rDNA sequencing method is relatively expensive, time- and labor-consuming, and not suitable for routine identification [11]; however, it is considered to be a decisive classification technique. The combination of molecular and spectrometric methods can have a large potential in complex taxonomic studies of bacteria [12]. Especially in the case of closely related taxa, like *Lactobacillus*, the identification at a subspecies level should be assessed by the sequencing of protein-coding genes, like *pheS*, which have a higher discriminating power than the 16S rRNA [13].

In recent years, the MALDI-TOF MS technique (matrix-assisted laser desorption ionization time-of-flight mass spectrometry) has become a promising and reliable tool for bacterial identification [9,10,14,15]. It is a rapid, simple and cost-effective method. In the food production sector, pharmaceutical sector and hospitals, where ensuring product safety is crucial due to consumer health concerns, there is a need to develop/adopt rapid and reliable methods of product control with regard to biological hazard. However, the applicability of the MALDI-TOF MS method to identify LAB species originating from food and beverages is still limited. Because of the large and still growing number of known species belonging to this genus and because of their biochemical and genetic diversity [10,14], the taxonomic classification of lactobacilli is not an easy task. The MALDI-TOF MS technique has several limitations in differentiating bacterial species. Other studies have shown that the different experimental approaches, media, times of cultivation and the sample preparation methods or the matrices used affect the obtained mass spectra and some of the differences in closely related species and strains can be very subtle [16,17]. The MALDI-TOF MS method is based on the comparison of the peptide spectroscopic fingerprints of the analyzed materials with the available data in the database. For new strains/isolates that have not been previously fingerprinted, identification is not possible or is unreliable. Raw cultures or cell extracts are commonly used for MALDI-TOF MS microbial analysis. The chemicals present in samples are ionized to produce charged molecules, whose mass-to-charge ratio could be measured. The mass spectra generated by ribosomal proteins is assumed to be a characteristic of each bacterial species. Nevertheless, the well created and updated database is the key.

Because the cellular perception and impact of HHP on cells is largely related to effects induced on proteins [1], this pressure treatment might cause significant changes in bacterial proteome. The present study aimed to evaluate the possible effect of HHP treatment on protein-based identification by MALDI-TOF MS. The study framework also includes the assessment of complementarity of the results produced by MALDI-TOF MS with those obtained by other molecular identification methods, namely the 16S rDNA sequencing.

## 2. Materials and Methods

### 2.1. Bacteria and Growth Conditions

Thirteen strains of LAB were used in this study, of which nine strains were isolated from spoiled food (juices and beers) samples commercially obtained from clients, to determine the source of the microbial contamination of the products. Two strains were isolated from bread and diet supplements. Isolation was performed according to the ISO 15214:2000 method using MRS agar (*Lactobacillus* Agar according to DeMan, Rogosa and Sharpe, Merck KGaA, Darmstadt, Germany) and incubation at 30 °C for 72 h. Additionally, one reference strain isolated from spoiled beer and collected from Leibniz Institute -DSMZ German Collection of Microorganisms and Cell Cultures ((Braunschweig, Germany)was used.

The strains were cultured in MRS broth medium at 30 °C for 48–72 h under anaerobic conditions. The growth kinetics of the strains were assessed in the early stationary phase by optical density measurements OD_600_ (Densitometer DEN-1B, BioSan, Riga, Latvia).

### 2.2. Isolation of Bacterial DNA

DNA from an overnight bacterial culture (1.5 mL) was isolated using the ExtractMe DNA Bacteria Kit (Blirt SA–DNA Gdansk, Poland) according to the manufacturer’s instructions. Briefly, the centrifuged bacterial pellets were suspended in 300 µL of BacL Buffer and supplied with 4 µL of RNase A, mixed, and incubated at 37 °C for 10 min. Then, 10 µL of Proteinase K was added to each sample and vortexed (Multi-Vortex V-32, BioSan, Riga, Latvia), and the mixture was incubated for 10 min at 55 °C. Subsequently, the samples were incubated for 5 min at 55 °C with BacB Buffer. After incubation, the samples were vortexed for 15 s and centrifuged for 2 min at 12,000 rpm (MiniSpin plus, Eppendorf AG, Hamburg, Germany). The supernatants were transferred into purification minicolumns in collection tubes and centrifuged once again for 1 min at 12,000 rpm. The purification minicolumns were then washed twice with BacW Buffer (600 µL and 400 µL, respectively). The samples were then centrifuged to remove the buffer residues. Subsequently, the samples were added to new sterile Eppendorf tubes, and the elution was performed with the Elution Buffer pre-heated to 70 °C; the eluted samples were centrifuged for 1 min at 12,000 rpm. The purity and concentration of the nucleic acids were confirmed by gel electrophoresis and UV spectrophotometry. The isolated DNA was stored at −20 °C.

### 2.3. Genotypic Identification

The species affiliation of the LAB was defined on the basis of the 16S rDNA sequence analysis. The 16S rDNA gene from the *Lactobacillus* species was amplified using the specific primers designed to amplify almost the entire gene sequence (around 1500 bp). The PCR mixture and the cycle parameters were as follows: the total PCR mixture volume was 60 µL, which contained 30 µL of the Dream Taq PCR Master Mix (Thermo Scientific), 1 µL of each primer (16SF: 5′-AGACTTTGATCCTGGCTCAG-3′ and 16SR: 5′-ACGGCTACCTTGTTACGACT-3′) in the final concentration of 0.4 µM, and 10–20 ng of DNA. The amplification was run in peqSTAR 2X thermocycler (VWR, Peqlab), and the conditions were as follows: initial denaturation for 2 min at 94 °C, followed by 40 cycles of denaturation for 30 s at 94 °C, annealing for 35 s at 51 °C and elongation for 100 s at 72 °C, with a final elongation for 2 min at 72 °C. The size of the amplified PCR products was analyzed by agarose gel electrophoresis in 1.5% (*w/v*) gel. Electrophoresis was performed at 120 V for 45 min in a 1× TAE buffer (Tris-Acetate-EDTA buffer). The gel was stained with 6 µL SimplySafe (EURx, Gdansk, Poland) to visualize the amplified fragments under a UV source. Gel documentation was performed with a BioImaging Systems 06-2d.1-G: BOX (Syngene, UK). The obtained products of amplification were sequenced using a 96-capillary 3730xl DNA Analyzer (Applied Biosystems-Life Technologies). The nucleotide sequences were analyzed and compared to the RefSeq database of the NCBI BLAST (Basic Local Alignment Search Tool) program.

### 2.4. Application of HHP

The influence of HHP on the ability to identify LAB by MALDI-TOF was assessed. Cultures from the stationary growth phase were exposed to high pressure treatment using U 4000/65 apparatus (Unipress, Warsaw, Poland). The volume of the treatment chamber was 0.95 L with the maximum working pressure of 600 MPa. A mixture of distilled water and polypropylene glycol (1:1, *v/v*) was used as the pressure-transmitting fluid. The working temperature of the apparatus was in the range between −10 °C and 80 °C. Pressure up to 300 MPa was generated in 70–80 s, and the release time was 2–4 s. Four milliliters of each culture was subjected to HHP at 300 MPa at an ambient temperature (20 °C) and held for approximately 5 min. The pressurization times reported did not include the come-up and come-down times. For each tested strain, the assays were performed by two independent processes. After the treatment, the samples were stored at room temperature (RT) for further analysis. Unpressurized samples were used as a control.

### 2.5. Species Identification by MALDI-TOF MS

The obtained isolates were analyzed with the MALDI-TOF MS technique (Bruker Daltonik, Germany). The protein extract prepared with ethanol and formic acid was used for analysis. Briefly, a loop of bacteria culture from the MRS plate was diluted in 150 μL MQ water, and 450 μL of 99.8% ethanol (POCH, Poland) was then added and the sample was carefully vortexed in RT. Then, the sample was centrifuged for 5.5 min at 13,000 rpm in RT. The supernatant was removed, and the residue was mixed with 40 μL of 70% formic acid and 40 μL of acetonitrile (Fluka, analytical grade) and vortexed in RT. After the centrifugation (13,000 rpm for 2.5 min, in RT) of the sample, 1 μL of supernatant was placed on a steel plate (Ground Steel) and allowed to air-dry in RT. Then, 1 μL of matrix solution containing 10 mg/mL of HCCA (α-cyano-4-hydroxycinnamic acid) was added to the plate and allowed to dry in RT.

Mass spectra were analyzed using the MALDI-Biotyper 3.0 software package (Bruker Daltonik, Germany) containing 5291 reference spectra. The results were shown as the top 10 identification matches along with confidence score (ranging from 0.00 to 3.00). On the basis of the criteria proposed by the manufacturer, a score below 1.70 does not allow for reliable identification; a score between 1.70 and 1.99 allows identification up to the genus level; a score between 2.00 and 2.29 reflects probable identification at the species level; and a score higher than 2.30 (2.30–3.00) indicates the highly probable identification at the species level. Both the untreated and HHP-treated strains were tested; the strains were analyzed in two independent experiments and each sample was run three times.

The effect of the stress factor HHP was evaluated by the dendrogram of the identified unpressured and pressured strains by using BioNumerics 7.6.3. (Applied Maths). The preprocessed mass spectra were preprocessed by smoothing and normalizing signal intensities. The triplicates of mass spectra of all strains were combined in 26 mass spectrum profiles (MSPs) and used for cluster analysis. The alignment and clustering were performed with 300 ppm linear tolerance and 0.5 constant tolerance. Cluster analysis was performed with the UPGMA (unweighted pair group method with arithmetic mean) method to visualize the dynamics of stress response and especially to determine the grouping of the *Lactobacillus* strains based either on their species affiliation or on the HHP impact.

The two-dimensional hierarchical clustering analysis (HCA) of the bacterial isolates versus their *m/z* values was performed. The cluster and heatmap were created on the basis of the similarity in their MS spectral pattern among all the unpressured and pressured strains. The heatmap was created by generating a tree for both the strains and peak classes using the UPGMA algorithm based on Spearman rank correlations. In the 2D cluster, the strains were displayed as column entries and peak classes were represented as row entries. Two dendrograms were shown: one above the bacteria entries and the other one next to the *m/z* values. In the matrix panel, the intensity of the peaks was represented by colors. Green implied low intensities and red implied high intensities.

## 3. Results

### 3.1. Identification of Lactobacillus Strains by 16S rDNA

All the isolates identified through the 16S rDNA sequencing revealed 99–100% homology to the sequences of the reference *Lactobacillus* strains deposited in GenBank. The 13 tested *Lactobacillus* strains obtained from the spoiled food and beverages were identified as the *Lactobacillus brevis* strains (5) (including one from the DSMZ collection), *Lactobacillus plantarum* (2), *Lactobacillus rhamnosus* (2), *Lactobacillus backii* (2), *Lactobacillus curvatus* (1), and *Lactobacillus rossiae* (1) (Table 1). According to the latest nomenclature [18], the strains will be named accordingly: *Lactobacillus brevis*—*Levitlactobacillus brevis*, *Lactobacillus plantarum*—*Lactiplantibacillus plantarum*, *Lactobacillus rhamnosus*—*Lacticaseibacillus rhamnosus*, *Lactobacillus backii*—*Loigolactobacillus backii*, *Lactobacillus rossiae*—*Furfurilactobacillus rossiae* and *Lactobacillus curvatus*—*Latilactobacillus curvatus*. On the basis of the 16S rDNA sequencing results, the phylogenetic tree (Figure 1) was created using the neighbor-joining method with distances computed with the maximum composite likelihood method using MEGA7 [19]. As expected, strains belonging to the same *Lactobacillus* species clustered together, where the groups could be identified: the first one included all *L. brevis* (except one, 557), the second one grouped two *L. plantarum* strains, and the third group included two *L. rhamnosus* strains. Among the other strains, one *L. curvatus* strain and one *L. backii* strain showed high similarities to the two *L. rhamnosus* strains; the 738-*L. rossiae* strain was excluded from the main cluster and the 103-*L. backii* and 557-*L. brevis* strains formed another small cluster.

To compare the identification level made by MALDI-TOF MS to that achieved by the 16S rDNA sequencing, a dendrogram was created using the MSPs of all the unpressured strains (Figure 2). The dendrogram showed the relationships of similarity among the strains and clustered them in one main group that comprised almost all the strains, except four strains identified by the MALDI-TOF MS as: 738-*L. rhamnosus*, DSM 6235 (115)-*L. brevis*, 103-*L. brevis*, and 557-*L. curvatus*. At a distance level of 60, all the strains were separated into a single *Lactobacillus* strain; however, the clustering showed no similarity to the 16S rDNA phylogenetic tree.

### 3.2. Identification of Lactobacillus Strains with the MALDI-TOF MS Method and the Influence of HHP

Thirteen isolates were classified as bacteria of the genus *Lactobacillus* before and after pressurization. Probability scores varied for the tested samples: results in the range of 2.30–3.00 were obtained for 6 out of 13 (46%) unpressured strains (978, 863, DSM 6235 (115), 102, 103, 3/16/1) and 8 out of 13 (61.5%) pressured strains (863, 738, 489, 133, DSM, 102, 103, 3/16/2). The other strains (except for the one unpressured strain with the score value of <2.00 (432)) scored at the 2.00–2.29 level; this allowed us to classify them as highly probable identified at the genus level and probable identified at the species level (Table 1), which yielded 46% (6/13) unpressured and 38% (5/13) pressured strains, respectively.

To visualize the relationship between all unpressured and pressured strains, the hierarchical cluster analysis was performed on the basis of a similarity matrix calculated using the curve-based Pearson correlation coefficient (Figure 3). MSPs were created in relation to their mass signals and peak intensities. On the basis of the maximum dissimilarities level, the dendrogram clustered the strains into one main group, except five strains including one pressured 557 strain and four unpressured ones: 863, 738, DSM (115), and 103. The strain 432 was clustered into one group at the distance level of 60, regardless of the difference in species identification: the unpressured one was *L. brevis* and the pressured one was *L. rhamnosus*. The strain 489 together with the unpressured strain 863 was clustered at the distance level of 96, and the strain 1178 was grouped in one cluster at the distance level of 90.

### 3.3. Comparison of Genotypic and MALDI-TOF MS Identification of Lactobacilli

The comparison of the results of the genomic analysis and MALDI-TOF MS allowed us to correctly identify 38% of the unpressured strains (5/13) and 31% of the pressured strains (4/13). Among the five correctly identified unpressured strains, three had the score value in the 2.00–2.29 range and the remaining two in the 2.30–3.00 range; among the four correctly identified pressured strains, one strain showed a score value in the 2.00–2.29 range and the remaining three strains in the 2.30–3.00 range. The cluster analysis based on the protein profile did not group the strains corresponding to the identification based on the 16S rDNA phylogenetic tree (Figure 1) and did not allow us to group the strains based on the influence of the stress factor (untreated or treated with HHP) (Figure 3). The dendrogram based on the mass spectra of the samples clustered the strains into two main groups: one contained different strains identified by MALDI-TOF MS as *L. brevis* and the other with few *L. brevis* and *L. plantarum* strains. Few of the analyzed strains were excluded from the cluster: 863HHP-*L. brevis*, 738noHHP-*L. brevis*, DSM 6235 (115) noHHP-*L. brevis*, 103noHHP-*L. brevis*, and 557HHP and noHHP-*L. curvatus*).

To assess the similarities and dissimilarities in the data, multidimension scaling (MDS) was used and the resulting data positions are displayed by 3D visualization (Figure 4). However, the MDS did not show the grouping of the strains based on their affiliation or whether they were treated or not treated with HHP.

The influence of the stress factor HHP on the protein profiles can be seen especially in MSPs of 133 and 432 strains. The data obtained with the mMass software [20] from their protein mass fingerprinting analysis (Figure 5) showed differences in the unpressured and pressured MSPs. The different protein profiles correlated with different identifications for those strains. The strains identified as *L. rhamnosus* in 16S rDNA (133), *L. curvatus* (432) and *L*. *rossiae* in MALDI-TOF MS identification were respectively *L. plantarum, L. rhamnosus* and *L. rhamnosus* for the unpressured and *L. brevis* for all three pressured strains.

The 2D clustering analysis was performed (Figure 6), and the heatmap was generated for all the studied strains of *Lactobacillus* versus their *m/z* values. The heatmap represents the differences among the unpressured and pressured strains according to the similarity in their spectral patterns. The figure shows that no bias in the distribution of the spectra can be directly correlated with the impact of HHP, which suggests that the MALDI-TOF MS data were not preferentially clustered according to the stress conditions.

## 4. Discussion

In recent years, MALDI-TOF MS has become a promising and reliable tool for the identification of bacteria isolated from different food sources. As a rapid and low-cost technique, it allows to analyze multiple samples in a short period of time and is expected to yield high score value identification results. Rapid microbial identification is becoming extremely important in the food industry. However, the applicability of this method is limited, and the limitation emerges from the reference database that needs to be constantly updated and expanded to include new species.

In the present study we evaluated the usability of the MALDI-TOF MS technique as a new, faster and easier approach to identify *Lactobacillus* from spoiled food and beverages. The influence of HHP on the *Lactobacillus* strains allowed us to study how the MALDI-TOF MS identification is processed when it is based on the protein profile that can be affected by the HHP treatment. Considering that only scores > 2.00 for MALDI-TOF MS results are accepted for species-level identification and scores 1.70–2.00 are accepted only for genus level identification, our results are not convincing for all the studied strains; moreover, they do not concur with the results of the genomic analysis. The two methods showed divergence in the identification of some of the *Lactobacillus* strains (two strains of *L. brevis* and *L. backii*, and one strain each of *L. plantarum, L. rhamnosus, L. rossiae,* and *L. curvatus*). Difficulty in identification by MALDI-TOF MS was observed when the strains were treated with HHP, leading to significant differences in the results.

Different stress factors such as the HHP treatment, which is used in the food industry, are likely to play an important role in the identification of LAB by MALDI-TOF MS. It can be presumed that the changes that occur at the proteomic level during the HHP process can impact the identification results obtained by MALDI-TOF MS. The biggest challenge in the differentiation and identification of closely related species is the inadequate and limited size of the database and the lack of efficient software. It is necessary to expand the database with a wide range of mass spectra dedicated to the high-throughput study of microorganisms derived from different environments.

The discriminatory inability of MALDI-TOF MS to identify some of the *Lactobacillus* species, which were identified by 16S rDNA sequencing (such as *L. backii*), suggests that the validation of the MALDI-TOF MS output by using well known molecular techniques is still required. Further studies with a higher number of strains, different pressure levels and proteomic effects, as well as extending the database, are needed to better assess the protein-based characterization techniques with the ability to identify species within the *Lactobacillus* genus.

In conclusion, our studies did not reveal the high congruence of LAB species identification by MALDI-TOF MS and 16S rDNA sequencing analysis. We provided important information on *Lactobacillus* isolated from spoiled food and beverages. Our findings indicate that the HHP treatment can affect the results obtained by MALDI-TOF MS. It is important to consider that the stress factors, cultivation methods and the natural environment from which bacteria are derived can affect their protein profile and thus change the mass spectrum, leading to differences in identification outcome.

The MALDI-TOF MS identification method needs additional use of genotyping; however, the choice of a suitable method to identify LAB strains should consider not only the discriminatory power of the techniques but also its cost-and time-efficiency.

## Figures and Tables

**Figure 1 microorganisms-08-00813-f001:**
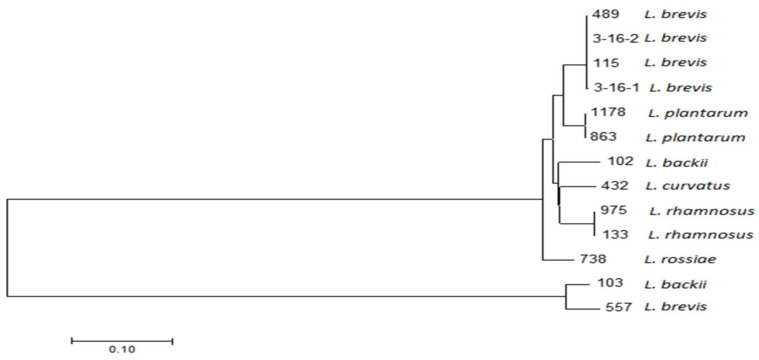
Phylogenetic tree based on the 16S rDNA sequences of the studied *Lactobacillus* strains. The neighbor-joining method was used to compute the evolutionary distances. The tree is drawn to scale, with the branch lengths measured in the number of substitutions per site. The tree was obtained using the MEGA7 software.

**Figure 2 microorganisms-08-00813-f002:**
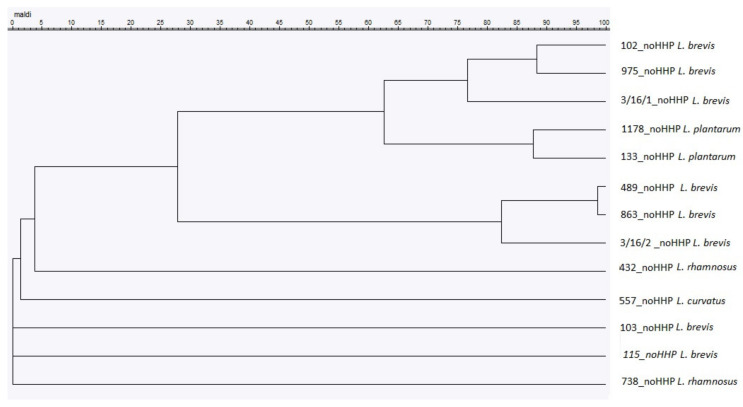
Cluster analysis of the MALDI-TOF MS spectra obtained from the *Lactobacillus* strains included in this study. In the MSP dendrogram, the relative distance between the strains is displayed as arbitrary units, where 100 indicates complete similarity and 0 indicates maximum dissimilarity.

**Figure 3 microorganisms-08-00813-f003:**
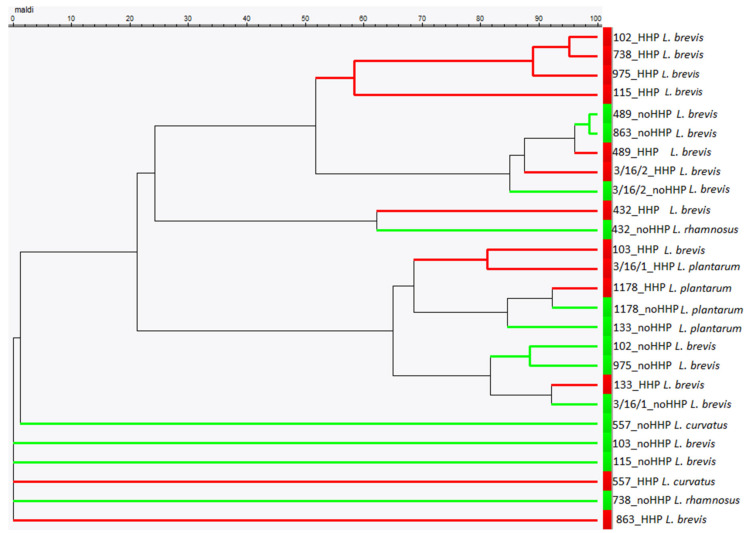
Dendrogram of the composite mass spectra illustrating the hierarchical clustering of the 13 unpressured (green-noHHP) and 13 pressured (red-HHP) strains identified by MALDI-TOF MS (names of strains according to MALDI-TOF MS identification). In the mass spectrum profile (MSP) dendrogram, the relative distance between the strains is displayed as arbitrary units, where 100 indicates complete similarity and 0 indicates maximum dissimilarity.

**Figure 4 microorganisms-08-00813-f004:**
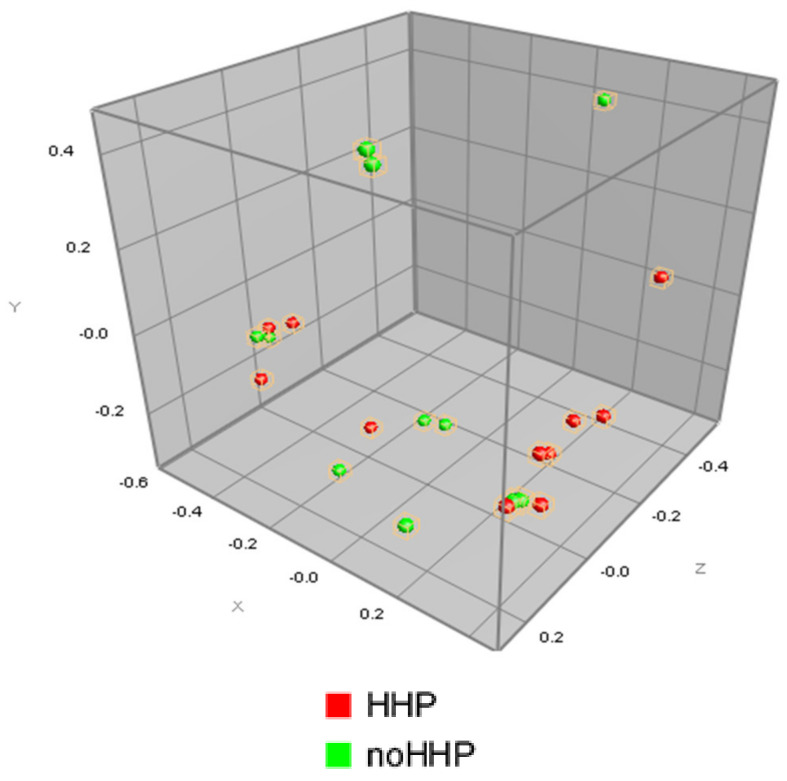
Multidimensional scaling representation of the mass spectra of the 13 strains of *Lactobacillus* obtained from the unpressured (noHHP -green) and pressured (HHP -red) strains.

**Figure 5 microorganisms-08-00813-f005:**
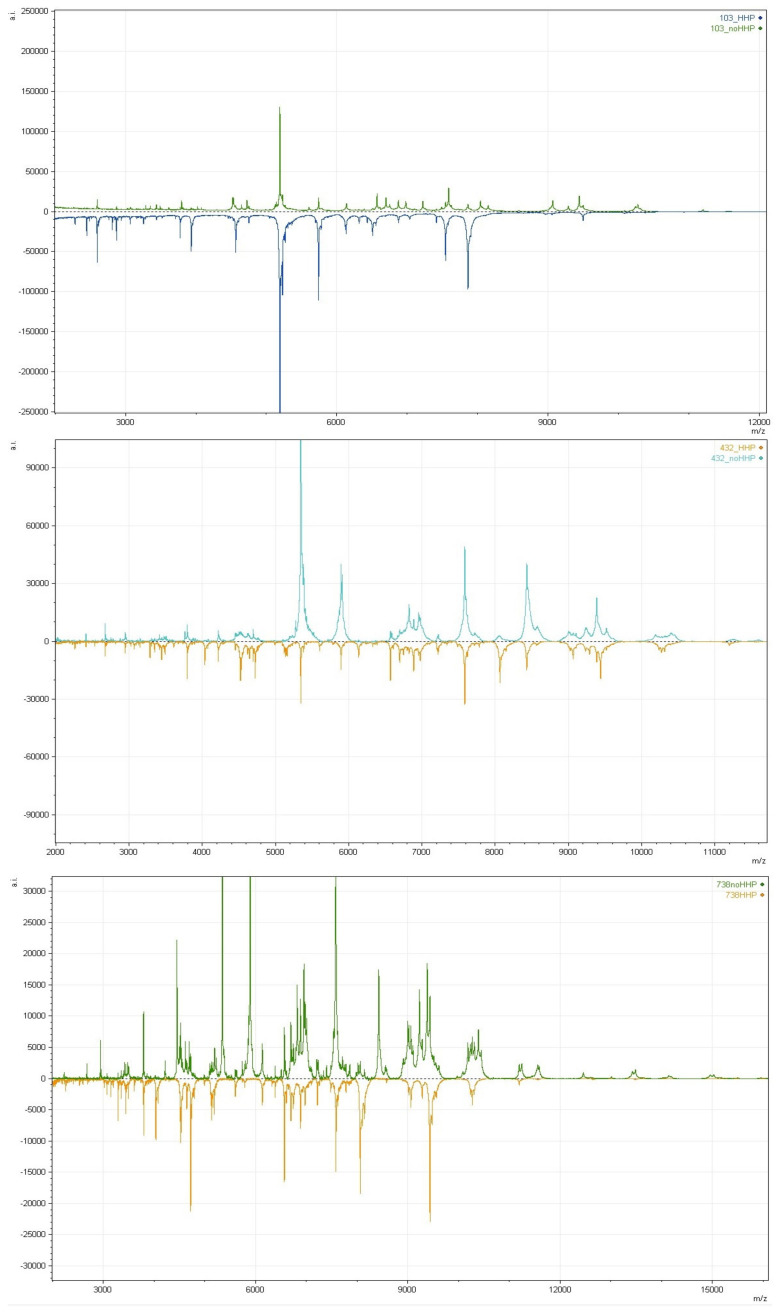
MALDI-TOF MS mass spectra obtained from the *Lactobacillus* strains (133, 432 and 738) that yielded different MPS identification results for the unpressured (noHHP) and pressured with 300 MPa/5′ (HHP) strains.

**Figure 6 microorganisms-08-00813-f006:**
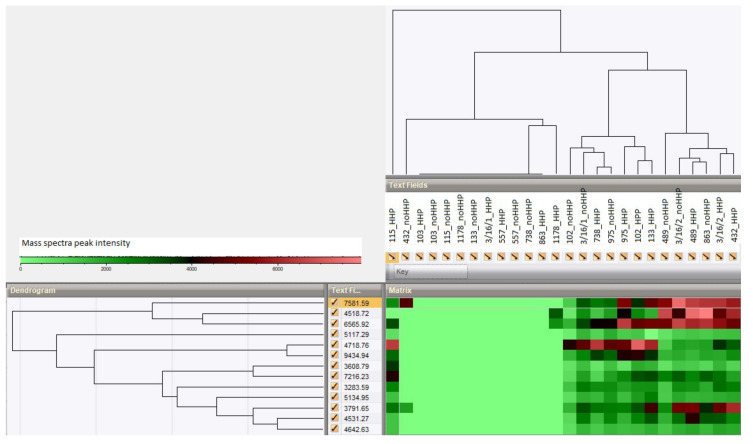
Two-dimensional hierarchical clustering analysis of the unpressured and pressured lactobacilli strains and the *m/z* values of the peak-matching mass spectra. The green indicates low peak intensity and the red indicates high peak intensity.

**Table 1 microorganisms-08-00813-t001:** Comparison of the results obtained for the unpressured and pressured samples by using the 16S rDNA sequencing and MALDI-TOF MS.

No.	Strain Number	Source	16S rDNA Sequencing Unpressured Strains	MALDI-TOF MS Unpressured Strains	Score Value	MALDI-TOF MS Pressured 300 MPa/5′ Strains	Score Value
1.	1178	Bread	*L. plantarum* (99%)	*L. plantarum*	2.04	*L. plantarum*	2.10
2.	975	Tomato juice	*L. rhamnosus* (99%)	*L. brevis*	2.36	*L. brevis*	2.28
3.	863	Tomato juice	*L. plantarum* (100%)	*L. brevis*	2.35	*L. brevis*	2.35
4.	738	Beer	*L. rossiae* (99%)	*L. rhamnosus*	2.05	*L. brevis*	2.33
5.	557	Beer	*L. brevis* (100%)	*L. curvatus*	2.00	*L. curvatus*	2.02
6.	489	Beer	*L. brevis* (99%)	*L. brevis*	2.25	*L. brevis*	2.34
7.	432	Saurkraft juice	*L. curvatus* (99%)	*L. rhamnosus*	1.95	*L. brevis*	2.23
8.	133	Probiotic	*L. rhamnosus* (99%)	*L. plantarum*	2.15	*L. brevis*	2.33
9.	DSM 6235 (115)	Beer	*L. brevis* (99%)	*L. brevis*	2.34	*L. brevis*	2.38
10.	102	Beer	*L. backii* (99%)	*L. brevis*	2.33	*L. brevis*	2.65
11.	103	Beer	*L. backii* (99%)	*L. brevis*	2.32	*L. brevis*	2.32
12.	3/16/1	Beer	*L. brevis* (99%)	*L. brevis*	2.33	*L. plantarum*	2.21
13.	3/16/2	Beer	*L. brevis* (99%)	*L. brevis*	2.27	*L. brevis*	2.35

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
