# Peer review of "Influence of High Hydrostatic Pressure on the Identification of Lactobacillus by MALDI-TOF MS- Preliminary Study"

_microorganisms, 2020, doi:10.3390/microorganisms8060813_

Round 1

Reviewer 1 Report

Dear authors

The manuscript presents an interesting work about employment of MALDI-TOF in food industry. This technique has several application and quick identification of microorganisms is one the most used due to its realibility. This characteristic is based on the employment of similar initial condition from biological material. In this way, the showed results present the posibility to build a database of MALDI-TOF profile for HHP treatments in order to solve this problems.

However, some points in the manuscript have to been solve or clarifiyed:

1-Nomenclature of Lactobacillus genus has been reordered in last months. Please check the names following the nomenclature of the following article.

Zheng, J., Wittouck, S., Salvetti, E., Franz, C. M., Harris, H. M., Mattarelli, P., ... & Watanabe, K. (2020). A taxonomic note on the genus Lactobacillus: Description of 23 novel genera, emended description of the genus Lactobacillus Beijerinck 1901, and union of Lactobacillaceae and Leuconostocaceae. International Journal of Systematic and Evolutionary Microbiology, ijsem004107.  

2- In this way, to carry on with a profitable identification, all the strains should be identifyed based on pheS gene phylogenic. This housekeeping gene is the reference gene to differenciate lactic acid bacteria at species level. The secuenque and the methodology could be found in the following reference. Sánchez-Juanes, F., Teixeira-Martín, V., González-Buitrago, J. M., Velázquez, E., & Flores-Félix, J. D. (2020). Identification of Species and Subspecies of Lactic Acid Bacteria Present in Spanish Cheeses Type “Torta” by MALDI-TOF MS and pheS gene Analyses. Microorganisms, 8(2), 301.  

3- ¿are the bacteria viable after HHP treatment? I suppose not. In addition, the use of genetic characterization throught RAPD-M13 profile should be recommended to ensure the similarity between both analyzed cultures.  

4-Multidimensional scanning reveals an erratical behavior of MALDI-TOF profile but only one sample of each treatment have been analyzed. Please increase the number of sample to ensure the repeatabylity of the results. This point is mandatory to present conclusion based on solid results.  

5-Name of the strain in figure should be presented in different format to ease the visualization of results. Treatment indication could be after or before the species name, but the name of the strain should be put after the species name for example: "Lactobacillus plantarum 1178 noHHP" or "no_HHP Lactobacillus plantarum 1178". Also, species name will be write in italic and other information without italic.

Please check all figures.  

Best regards    

Author Response

Dear reviewers,

Thank you very much for the valuable suggestions and comments of our manuscript. On behalf of my co-authors, we would like to thank you for giving us an opportunity to revise our manuscript and giving us a chance for further improvement. We appreciate editor and reviewers very much for their positive and constructive comments and suggestions on our manuscript entitled „Influence of high hydrostatic pressure on the identification of Lactobacillus by MALDI-TOF MS”.

We have revised our manuscript carefully according to these suggestions and comments and point-by-point basis. I am glad to answer them and looking forward to yours answer. We have answered all questions you raised. The answers to the suggestions were marked in red and the revisions of the manuscript were highlighted in red. Other minor mistakes were also being revised and highlighted in red. If you have any more suggestions or questions, please do not hesitate to tell us.

Best wishes.

With regards,

Joanna Bucka-Kolendo

Reviewer 2 Report

  1. Affiliation: Please check carefully. Should be “street” in the address? Or “SokoÅ‚owska”?
  2. Please keep the completely name of “MALDI-TOF MS” in the all manuscript.
  3. Please put a shortened strain names in all manuscript, also in the Figures 1, 2. In the Figure 3 some of the names should have a space in the names for example “ brevis
  4. Line 35: “[2-5]”
  5. Line 91: “ºC”
  6. Line 99: “Gdansk”
  7. Line 136: “of each culture”
  8. Line 137: “20ºC”
  9. How the Author can explain that two methods have shown mainly the different results on unpressured strains? 5/13 strains
  10. Methods: Is the bacteria strains were identified by MALDI-TOF MS Biotyper platform? In method section is an information about protein extracts which were analyzed by MALDI-TOF MS method and then the received mass spectra were analyzed by MALDI-Biotyper 3.0 software package?
  11. Methods: Four milliliters of each bacteria culture was subjected to HHP and these cultures were stored at room temperature for further analysis. Please explain what does mean that small amount of bacterial colony was diluted in mQ ? (line 145)
  12. Methods: Please put the temperature values during all centrifuges and composition (solvent) of matrix solution; please put the standard information which was used.
  13. Line 144: “pure” replace with “percentage”
  14. Lines 150/151: Please remove the last sentence.
  15. Line 156: If the score between 1.70-1.99 allows identification up to the genus level why in the table 1 the strain 432 has been named rhamnosus (at score 1.95)?
  16. Methods: Is each protein extraction was repeated or only each sample with extracted protein was run for three times?
  17. Table 1. Please put a shortened strain names in the table. This will improve the appearance of the table. In the table description please put “.”; and in the table the score value of 1178 with two decimal places.
  18. Lines 169, 174, 262: “m/z
  19. Line 182: “(Table 1)”
  20. DSMZ or DSM – please check in the manuscript
  21. Line 201: “16S rDNA”
  22. Line 216: what does mean “high probable identified at the genus...”? The score values 2.00- 2.29 enabled identification at the genus level and reflects probable identification at the species level (fragment from earlier manuscript section)
  23. Line 255: with the strain 738 is the same situation. Why the Authors did not mention about 738 strain? Please remember that score 1.95 for 432 strain shows identification only on genus level.
  24. Figure 5. The figure is too large.
  25. Line 260: “.”
  26. Line 310: “higher number”
  27. In the discussion section please keep the values with two decimal places
  28. The References please prepare according to Instructions to Authors.

Author Response

(The authors gave the same response as above.)

Reviewer 3 Report

Despite the careful clustering analysis made by the authors, the low number of samples used does not allow an effective evaluation of the use of MALDI TOF for the identification of spoilage lactobacilli strains, nor the evaluation of the effect of HHP on the variation of the protein pattern of treated strains, therefore we think that the work does not bring any relevant indication to the scientif community.

Author Response

Dear reviewers,

Thank you very much for the valuable suggestions and comments of our manuscript. On behalf of my co-authors, we would like to thank you for giving us an opportunity to revise our manuscript and giving us a chance for further improvement. We appreciate editor and reviewers very much for their positive and constructive comments and suggestions on our manuscript entitled „Influence of high hydrostatic pressure on the identification of Lactobacillus by MALDI-TOF MS”.

We have revised our manuscript carefully according to these suggestions and comments and point-by-point basis. I am glad to answer them and looking forward to yours answer. We have answered all questions you raised. The answers to the suggestions were marked in red and the revisions of the manuscript were highlighted in red. Other minor mistakes were also being revised and highlighted in red. If you have any more suggestions or questions, please do not hesitate to tell us.

Point 1: Despite the careful clustering analysis made by the authors, the low number of samples used does not allow an effective evaluation of the use of MALDI TOF for the identification of spoilage lactobacilli strains, nor the evaluation of the effect of HHP on the variation of the protein pattern of treated strains, therefore we think that the work does not bring any relevant indication to the scientific community.

 Response 1: Thank you for the suggestion and we will take it into account in future research. In the current pandemic, it is not possible to repeat the studies in such a short time. We added “preliminary study" to the title, to mark that this is only a beginning of the work, as well in the discussion stands: “Further studies with higher number of strains, different pressure levels, and proteomic effects, as well as extending the database, are needed”.

As some proteins in bacteria undergo induction as a result of HHP and thus we wanted to evaluate how/if the HHP affects the identifications method which is based on the comparison of ribosomal proteins, such as MALDI-TOF MS. Therefor we believe that our work shows the value for the scientific community.

Best wishes.

With regards,

Joanna Bucka-Kolendo

Round 2

Reviewer 1 Report

Dear authors,

congratulations, the manuscript presents the quality to be published.

Best regards
